# Epstein-Barr Virus (EBV) Is Mostly Latent and Clonal in Angioimmunoblastic T Cell Lymphoma (AITL)

**DOI:** 10.3390/cancers14122899

**Published:** 2022-06-12

**Authors:** Racha Bahri, François Boyer, Mohamad Adnan Halabi, Alain Chaunavel, Jean Feuillard, Arnaud Jaccard, Sylvie Ranger-Rogez

**Affiliations:** 1Microbiology Department, Faculty of Pharmacy, 87025 Limoges, France; racha_bahri91@outlook.com (R.B.); mohamadadnan.halabi.1@ul.edu.lb (M.A.H.); 2Unité Mixte de Recherche CNRS (Centre National de la Recherche Scientifique) 7276 and INSERM (Institut National de la Santé et de la Recherche Médicale) U1262, 87025 Limoges, France; francois.boyer@unilim.fr (F.B.); jean.feuillard@unilim.fr (J.F.); arnaud.jaccard@chu-limoges.fr (A.J.); 3Pathology Department, University Hospital Dupuytren, 87000 Limoges, France; alain.chaunavel@chu-limoges.fr; 4Biological Hematology Department University Hospital Dupuytren, 87000 Limoges, France; 5Clinical Hematology Department, University Hospital Dupuytren, 87042 Limoges, France; 6Virology Department, University Hospital Dupuytren, 87000 Limoges, France

**Keywords:** angioimmunoblastic T cell lymphoma, AITL, Epstein–Barr virus, EBV, clonal, latent, NGS

## Abstract

**Simple Summary:**

Angioimmunoblastic T cell lymphoma (AITL) is the most common peripheral T cell lymphoma encountered in Europe. It is a non-Hodgkin’s lymphoma with a poor prognosis. The Epstein-Barr virus (EBV) virus is detected in more than 90% of biopsies, especially in large B lymphocytes. To date, the role of EBV in this pathology is still debated. The aim of our study was to analyze whole viral genomes in AITL compared to other EBV-associated lymphomas. We observed that two viral strains were mainly found in AITL, one of which appeared to be associated with poor post-diagnosis survival. Furthermore, the virus was found to be clonal and latent in all cases of AITL; for one biopsy, the virus was both latent and most likely replicative, depending on the cells. On the whole, these results support a role for EBV in AITL.

**Abstract:**

The Epstein-Barr virus (EBV) is associated with angioimmunoblastic T cell lymphoma (AITL), a peripheral T lymphoma of poor prognosis in at least 90% of cases. The role of EBV in this pathology is unknown. Using next-generation sequencing, we sequenced the entire EBV genome in biopsies from 18 patients with AITL, 16 patients with another EBV-associated lymphoma, and 2 controls. We chose an EBV target capture method, given the high specificity of this technique, followed by a second capture to increase sensitivity. We identified two main viral strains in AITL, one of them associated with the mutations *BNRF1* S542N and *BZLF1* A206S and with mutations in the *EBNA-3* and *LMP-2* genes. This strain was characterized in patients with short post-diagnosis survival. The main mutations found during AITL on the most mutated latency or tegument genes were identified and discussed. We showed that the virus was clonal in all the AITL samples, suggesting that it may be involved in this pathology. Additionally, EBV was latent in all the AITL samples; for one sample only, the virus was found to be latent and probably replicative, depending on the cells. These various elements support the role of EBV in AITL.

## 1. Introduction

The Epstein-Barr virus (EBV) is a ubiquitous human γ-herpesvirus that infects more than 95% of the adult population. EBV primary infection corresponds to the infection of the oropharyngeal epithelial cells, where it actively replicates, and of B cells, where it remains as a lifelong latent infection, without production of new virions. Nine latent proteins (Epstein-Barr Nuclear Antigens, EBNA-1, -2, -3A, -3B, -3C, and –LP, and Latent Membrane Proteins, LMP-1, -2A, and -2B) are encoded in the unique regions of the 172 kb EBV genome. Four different latency programs can be identified in latently infected cells, based on which the latency proteins are expressed. The EBV genome also encodes for the non-coding small microRNAs (miRNAs) that map to the BHRF1 or BART (BamHI A rightward transcripts) regions, found during latency, and for the non-coding small RNAs, named EBV-encoded small RNA-1 and -2 (EBER-1 and EBER-2), which are highly abundant during all forms of EBV latency. The EBNA-1 protein is expressed in all forms of latency and is found in all EBV-associated malignancies. It is a homodimer and plays an essential role in maintaining the latency state and retaining the viral episome in latently infected cells [1]. The latency protein EBNA-2 acts principally as a transcription factor, whereas EBNA-LP, expressed at the same time as EBNA-2, is principally a coactivator of the transcriptional activator EBNA-2.

Reactivation, which corresponds to the resumption of the lytic cycle with infectious virus production, may occur in the context of the activation of latently infected cells. It begins with the activation of the very early transactivators BZLF1 and BRLF1, which are responsible for the expression of very early genes, followed by early then late gene expression, resulting in the sequential expression of more than 80 genes. Different genomic variations have been reported. Based on variations in the latent genes, principally in *EBNA-2*, *-3A*, *-3B*, and *-3C* genes, two types of EBV may be identified [2]: EBV type 1 (EBV-1) and EBV type 2 (EBV-2). Whereas EBV-1 is more prevalent in the developed world, EBV-2 is principally encountered in equatorial Africa. EBV1, because it transforms B cells more efficiently than EBV2 and is found more often in lymphomas, is considered to be more aggressive than EBV2 [3].

EBV plays an important role in human diseases. Primary infection, which often occurs in childhood, is commonly asymptomatic in children but may cause infectious mononucleosis (IM) when it occurs later, principally in Western countries. Moreover, EBV is involved in the development of malignancies of a lymphoid origin, where it is, respectively, detected in 80% of endemic Burkitt’s lymphoma (BL) cases, 71% of nasal NK/T lymphoma (NK/TL) cases, in about 30% of Hodgkin’s lymphomas (HL), and more rarely in B- or T-cell lymphoproliferations in immunocompromised patients. It is also associated with epithelial malignancies, such as undifferentiated nasopharyngeal carcinoma (NPC), where it is always found and seems to be implicated in 10% of gastric carcinomas (GC). Intriguingly, the incidence of pathologies involving EBV varies enormously across geographic regions [4]. For example, endemic BL is essentially found in tropical sub-Saharan African countries, where it may account for approximately half of all childhood cancers [5], while for NPC, the highest incidence and mortality rates are observed in Southeast Asia [6]. In addition, a recent study showed that, in immunocompromised patients, different EBV strains are able to induce the development of lymphoid tumors with variable efficacy. These different observations plus the fact that EBV genome variations have been identified among individuals with various types of cancers [7,8,9] lead to the hypothesis of a potentially disease-specific EBV strain [4,10,11,12,13].

Angioimmunoblastic T cell lymphoma (AITL) is one of the most common forms of nodal peripheral T cell lymphoma (PTCL) [14], but it represents a small percentage (1–2%) of all cases of non-Hodgkin’s lymphomas (NHL). It is less common in North America or Asia than in Europe, the prevalence being, respectively, 16–18% and 29% [15,16]. No racial predisposition or risk factors have been found. AITL mostly affects advanced-age people with a median of ≥62 years of age at diagnosis and has an equal male to female ratio. Clinically, AITL is characterized by a multitude of peculiar presentations with non-specific symptoms, and diagnosis is often made late. The main clinical presentation is a generalized lymphadenopathy, often accompanied by B-symptoms, and more rarely hepatosplenomegaly or skin rash [17]. These symptoms can be associated with immunologic abnormalities such as polyclonal hypergammaglobulinemia or Coombs positive hemolytic anemia [18]. Pathologic findings show collapsed lymph node architecture, including a polymorphous infiltrate of eosinophils and plasma cells, large immunoblasts, B cells, histiocytes, epithelioid cells, and atypical T cells with abundant clear cytoplasm. The perivascular expansion of follicular dendritic cells (FDCs) and abundant arborizing endothelial venules are observed. Neoplastic cells, often less abundant than the reactive background, are localized in close proximity to the endothelial venules. Molecular analysis revealed that the cell of origin belongs to an effector T-cell subset, the follicular T helper (T_FH_) cell, which plays a key role in B-cell activation and differentiation in the germinal center [19,20].

No characteristic karyotypic abnormality has been found in AITL. Gene expression profiling identified many aberrations, which unfortunately were not sufficient to explain lymphomagenesis. Genotype studies reported the clonal rearrangement of *TCR* genes in up to 80% of cases and *IGH* gene rearrangement in up to one third of cases [21,22]. Genome-wide sequencing revealed a recurrent somatic mutation G17V in the small GTPase ras homology family member A (RHOA) in 50 to 70% of AITL cases [23], together with genetic alterations in the epigenetic modifier genes, principally *tet methylcytosine dioxygenase 2* (*TET2*) and, more rarely, *DNA methyltransferase 3 alpha* (*DNMT3A*) or *isocitrate dehydrogenase 2* (*NADP+*), mitochondrial (*IDH2*), and signaling factors (e.g., *FYN* and *CD28*) [24,25,26]. The overall prognosis of AITL is poor; the 5-year median survival is only 32% [27,28,29].

EBV positive cells are detected in up to 85–95% of AITL biopsies, the virus being principally located in large B-cell blasts [30,31,32,33,34], which may resemble Hodgkin’s RS cells and are distributed throughout tissues [35]. In some cases, the neoplastic T cells may also be infected. The role of EBV in AITL is still uncertain and several hypotheses coexist. Some authors argue that EBV reactivation occurs as a consequence of the immunodeficient state created by AITL, thereby favoring the expansion of T_FH_ and B cells and playing a role in the development of the tumor microenvironment. Others allege that EBV itself drives the development of AITL by activating T_FH_ cells [36]. The presence of EBV positive cells detected early in the disease course and the fact that EBV-positive B-cell proliferation may occur during AITL progression, imply that this virus may play a role in the development of AITL [31].

Here, using-next generation sequencing (NGS), we characterized the viral genomic alterations in EBV-associated AITL. For these patients, we described the mutations on the most mutated latent and tegument genes, and we demonstrated that EBV was in a latent state in all the AITL tumor biopsy samples and in a clonal form in all samples, which is consistent with a direct role for EBV in AITL.

## 2. Materials and Methods

### 2.1. Cell Lines and Culture

Seven EBV positive cell lines were used principally for methodology validation by a comparison of the obtained sequences to the published sequences, when available. They were also used as controls to detect viral clonality. The prototypical B95-8 strain served as a reference for the type 1 virus. Four Burkitt lymphoma cell lines (Jijoye, Namalwa, P3HR1, and Raji) were chosen for their different characteristics. All cell lines are available at ATCC (catalog numbers CRL 1612—ECACC 85011419, CCL-87, CRL-1432, HTB-62, and CCL-86, respectively, Manassas, VA, USA). Two NK cell lines, SNK6 and MEC04, were obtained from Marion Travert (Inserm U955, Hôpital Henri Mondor, Créteil, France). The B cell lines were grown in RPMI-1640 with glutaMAX (ThermoFisher Scientific, Illkirch-Graffenstaden, France; catalog number 61870-010) supplemented with 10% (*v*/*v*) fetal bovine serum, (FBS; Eurobio Scientific, Les Ulis, France; catalog number CVFSVF00-0U) and 100 units/mL penicillin, 100 μg/mL streptomycin (Eurobio Scientific, Les Ulis, France), and 15 mM HEPES (pH 7.4) (Eurobio Scientific, Les Ulis, France) at 37 °C with 5% CO_2_. The NK cell lines were cultured under the same conditions in the presence of 100 U/mL interleukin-2 (Sigma-Aldrich, Saint-Quentin Fallavier, France; catalog number I7908).

### 2.2. Patients

This study was retrospectively conducted on 34 frozen EBV positive biopsies collected at the initial diagnosis from patients hospitalized at Limoges University Hospital, France, between 2000 and 2015. All the lymphoma cases were initially diagnosed after examination by a pathologist and reviewed independently by another pathologist using WHO criteria [14]. The median age was 64 ± 15.59 years, and the sex ratio was 0.49. Details of this patient cohort are provided in Table 1. Furthermore, one EBV-positive inflammatory reactive biopsy, belonging to a 73-year-old woman without hematological malignancy, and a serum collected from a 13-year-old boy hospitalized for symptomatic IM were studied as controls.

Informed consent was obtained from all the patients to analyze their samples, and the study was approved by the Ethics Committee of the Institutional Review Board.

### 2.3. EBER In Situ Hybridization

In order select EBV-positive AIL samples, the EBV was detected by in situ hybridization (ISH). The formalin-fixed paraffin-embedded (FFPE) tissue sections used for this detection were initially deparaffinized, then rehydrated in a graded solution of xylene and alcohol and deproteinized with proteinase K (ThermoFisher Scientific, Illkirch-Graffenstaden, France). They were subsequently incubated with the Ventana EBER 1 DNP Probe^®^ (Roche Diagnostics, Meylan, France; catalog number 760-1209) used for EBER hybridization, followed by staining with the Ventana ISH iVIEW blue detection kit^®^ (Roche Diagnostics; catalog number 760-097). Images of the EBER staining obtained for control 1 (absence of EBER detection) and for a patient with AITL are visible in the Appendix A.

### 2.4. B and T Clonality Determination

B-cell clonality was evaluated according to the van Dongen publication [37] after amplification of the VDJ region in heavy chains, using the consensus IGHJ primer and IGHV primer families (FR1, FR2, or FR3), as well as in light chains, using the IGkJ and IGkV primer families following the amplification protocol described by the authors. TCR clonality was determined by TCRβ and TCRγ gene amplification using, respectively, Vβ or Dβ family primers and the Jβ primer family for TCRβ or the Vγ and Jγ primer family) for TCRγ. All the primers were purchased from Sigma-Aldrich, Saint-Quentin Fallavier, France. The PCR products obtained from the Ig and TCR gene rearrangements were analyzed by heteroduplex and GeneScanning analysis (Applied Biosystmes, ThermoFisher Scientific, Illkirch-Graffenstaden, France). For heteroduplex analysis, PCR products obtained with unlabeled primers (Sigma-Aldrich) were denatured at a high temperature (95 °C for 5 min), followed by low-temperature rapid random renaturation (4 °C for 1 h). The products were then submitted to electrophoresis on a 6% polyacrylamide gel to distinguish between homo- and heteroduplexes. For GeneScanning analysis, the fluorochrome-labeled single-strand (denatured) PCR products were size separated in a denaturing polyacrylamide sequencing gel or a capillary sequencing polymer (Applied Biosystmes) and detected via automated scanning with a laser (Applied Biosystmes).

### 2.5. DNA Extractions and Generic Amplification of Serum DNA

DNA was extracted from the cell lines (10^6^ cells) and frozen tissue samples (6 sections of 10 μm each) using the DNeasy Blood and Tissue Kit^®^ (Qiagen, Les Ulis, France; catalog no. 69504) according to the manufacturer’s recommendations. The DNA concentration was determined by the Qubit 2.0 Fluorometer^TM^ (Life Technologies, Villebon-sur-Yvette, France). DNA was extracted from the serum using the NucliSENS EasyMAG platform^TM^ (BioMérieux, Marcy-l’Etoile, France). Because the amount of viral DNA was too low in the serum samples, a generic amplification was conducted using the TruePrime WGA kit^®^ (Ozyme, Saint-Cyr-l’Ecole, France, catalog no SYG351025) according to the manufacturer’s instructions. This technique, based on multiple displacement amplification, uses two enzymes: the newly discovered primase DNA TthPrimPol and the highly processive Phi29 DNA polymerase.

### 2.6. Probe Design for EBV Sequence Capture

Full-length EBV genomes of EBV type1 (NC_007605) and EBV type2 (NC_009334) prototypes were used to design the EBV probes by Roche (Madison, WI, USA). Overlapping 100–120 mer DNA probes were designed so as to cover the EBV genomes at least five times. The coverage was estimated at almost 99.7% and 99.9% of the EBV1 and EBV2 genomes, respectively. The probes did not match to the human hg19 genome (GRch38.p13) as determined by the SSAHA algorithm. A probe was considered to match to the genome if there were less than five single-base insertions, deletions, or substitutions between the probe and the genome. The vast majority of probes were unique, with a few probes that had a greater degree of multi-locus homology to increase the coverage in all regions.

### 2.7. Illumina Library Construction and Whole EBV Genome Sequencing

The overall experiment was conducted according to the NimbleGen Seqcap EZ library SR manufacturer’s protocol^®^ (Roche). Two micrograms of each DNA sample were fragmented using the Bioruptor Sonicator^TM^ (Diagenode, Liège, Belgium) with a size range between 100 and 400 bp fragments, using the following settings: volume, 100 μL, temperature, 4 °C, number of cycles, 13, for the cell lines and frozen biopsies. The fragmented samples were subsequently used to synthetize libraries with the KAPA Library Preparation Kit^®^ for the Illumina NGS platform (KAPA Biosystems, Roche, catalog no 07137974001) according to the manufacturer’s recommendations. First, the fragments were submitted to the end-repair process to obtain blunt ends. Then, they were “A tailed” by the addition of nontemplated Adenine nucleobase, adaptor-ligated and index-tagged. The libraries were enriched with a PCR reaction for 8 cycles by ligated mediation PCR (pre LM-PCR). The final size selection of the library was achieved by a single AMPure XP Paramagnetic Bead^®^ (Agencourt, Beckman Coulter Genomics, Villepinte, France, catalog no A63881) cleanup, targeting a final 300 to 500 bp library size. The libraries underwent a qualitative (final size distribution) and quantitative assay using the High Sensitivity DNA Labchip kit^®^ (Agilent, Technologies, Les Ulis, France, catalog no 5067-626) on the 2100 Bioanalyzer (Agilent). The obtained libraries were pooled at equal molar quantities to a total of 1 microgram and hybridized with EBV biotinylated probes at 47 °C for 24 h. The hybridized fragments containing the targeted genes were adsorbed on magnetic streptavidin-beads, and the uncaptured DNA fragments were removed by washing. Enrichment of the eluted fragments was then performed by middle ligated mediation PCR (middle LMP-PCR) for 5 cycles. A second capture, a washing, and a post LM-PCR of 14 cycles were conducted to increase the hybridization yield, as AITL lymphomas are known to contain low copy numbers of EBV. The final concentration of each capture pool was verified through the 2100 Bioanalyzer, and sequencing was performed using Illumina technology with paired-end sequenced DNA libraries. According to the viral load before library synthesis, two Illumina sequencing platforms were used. The Illumina MiSeq Reagent Kit^®^ V2 (2 × 250 bp pair-end sequencing) and the Illumina Nextseq Reagent Kit^®^ (2 × 75 bp pair-end sequencing) (Illumina, ICM, Hôpital Pitié, Paris, France) were used, respectively, for samples with a viral load greater than and less than 100,000 copies/μg DNA. To validate the sequencing workflow, the MiSeq personal sequencer laser (Illumina, Evry, France) was used to resequence B95-8, Jijoye, P3HR1, and Raji whose sequences had been previously published.

### 2.8. Bioinformatics Analysis

The sequencing read pairs were first demultiplexed into fastq files using the MiSeq reporter (Illumina) by allowing one mismatch in the index sequence. The removing of the 3′/5′ adaptors was performed by using the Cutadapt tool implemented in the Trim Galore program (https://usegalaxy.org/?tool_id=toolshed.g2.bx.psu.edu%2Frepos%2Fbgruening%2Ftrim_galore%2Ftrim_galore%2F0.4.3.1&version=0.4.3.1&__identifer=77iq1zkls3t) accessed on 3 March 2021.

#### 2.8.1. De Novo Assembly

VirAmp, a galaxy-based viral genome assembly pipeline, was used to generate scaffolds from the assembly of sequencing read pairs with a de Bruijin graph algorithm. Briefly, after stringent quality control and host decontamination steps, the Velvet de novo assembler is called to generate contigs. From these contigs, the Viramp pipeline assembles longer scaffolds, summarizes assembly statistics, and displays the contigs’ alignment to the reference genome in a Circos graph [38]. The scaffolds generated by Viramp were oriented to the reference genomes (NC_007605 for type 1 or NC_009334 for type 2) with BLAST (https://blast.ncbi.nlm.nih.gov/Blast.cgi, accessed on 10 June 2021) [39], and linear genome sequences were built accordingly by resolving overlaps with 5′ priority and by filling the gaps with Ns.

#### 2.8.2. Read Mapping and Variant Calling

A variant calling pipeline was designed on the Galaxy main server [https://usegalaxy.org, accessed on 10 June 2021. Base quality control was performed with the sliding window trimming operation (number of base to average across: 4, average Phred quality required: 20 for NextSeq data, 25 for MiSeq data) from Trimmomatic (https://usegalaxy.org/?tool_id=toolshed.g2.bx.psu.edu%2Frepos%2Fpjbriggs%2Ftrimmomatic%2Ftrimmomatic%2F0.36.3&version=0.36.3&__identifer=21rzgzebycc, accessed on 9 March 2021 [40]. High-quality reads from each sample were then mapped to the human reference genome (NCBI build 37, HG19). Read pairs that aligned to the human genome were discarded while the unmapped reads were aligned to EBV reference genomes (NC_007605 and NC_009334). Both mapping steps were performed with Bowtie2 (https://usegalaxy.org/?tool_id=toolshed.g2.bx.psu.edu%2Frepos%2Fdevteam%2Fbowtie2%2Fbowtie2%2F2.3.2.2&version=2.3.2.2&__identifer=y4zlyajjmx, accessed on 10 March 2021 with default parameters [41]. Alignment statistics were retrieved with SAM tools [42], and position-wise read depths were computed with deepTools2 [43]. The resulting bam files were then converted to pileup with the Galaxy SAMtools wrapper “Generate pileup” (https://usegalaxy.org/?tool_id=toolshed.g2.bx.psu.edu%2Frepos%2Fdevteam%2Fsam_pileup%2Fsam_pileup%2F1.1.2&version=1.1.2&__identifer=ekt7zeimbgm, accessed on 16 March 2021). Nucleotide variants were called with VarScan (https://usegalaxy.org/?tool_id=toolshed.g2.bx.psu.edu%2Frepos%2Fdevteam%2Fvarscan_version_2%2Fvarscan%2F0.1&version=0.1&__identifer=8i8tfbb8q9l, accessed on 10 June 2021) with high-sensitivity parameters (min. read depth: 10, min. supporting reads: 5, min. variant allele freq.: 0.05, *p*-value: 0.99) [44] and reported in individual vcf files.

### 2.9. EBV Typing

As the type 1 and type 2 EBV sequences mostly diverge in *EBNA-2* and *EBNA-3* genes, the EBV type was determined by comparing the read mapping performance on these regions when the type 1 or type 2 sequences were used as a reference. For each sample, the ratio of the number of EBV1-mapped reads to the number of EBV2-mapped reads in the EBNA-2 and EBNA-3 regions was computed. A visual examination of read alignments was also performed in IGV Integrative Genomic viewers [45] to confirm the specificity of the read alignments over these regions.

### 2.10. Mutation Analysis and Clonality Assessment

The annotation of individual VCF files was performed using snpEff [46] with reference GenBank files NC_007605 (for type1) or NC_009334 (for type2). Variations found in major repeats (internal repeats and terminal repeats) were discarded. Custom Python scripts (available on request) were run to classify variants according to variation nature (synonymous or non-synonymous, substitution or insertion, or deletion), variant site homogeneity (heterogeneous if variant allele frequency is >5% or <95%), and affected protein function (according to the classification of Tabouriech et al., 2006) [47]. The proportion of heterogeneous sites was used to assess the clonality of each sample.

### 2.11. Phylogenetic Analysis of EBV Genomes

The reported homogeneous variations in the vcf files were parsed to generate whole genome or specific gene sequences for each sample. Multiple sequence alignments were obtained with MAFFT [48], and phylogenetic trees were built using the BioPython Phylo module functions to implement the UPGMA algorithm (Unweighted Pair Group Method with Arithmetic Mean).

### 2.12. Quantitative PCRs

The EBV viral load and albumin concentration were determined by TaqMan qPCR. The primer and probe sequences for EBV *BMRF1* and *human*
*albumin* gene amplification, designed by means of Primer Blast, were, respectively:

BMRF1s (forward), 5′-CCGGCCTGAATTTGTTAAGC-3′,

BMRF1as (reverse), 5′-CTTGGGCATCAACAGCACC-3′,

BMRF1p (probe), 5′-AATCATCTGCTCGTTCCTCAGCC-3′ and

AlbR (forward) 5′-AAACTCATGGGAGCTGCTGGTT-3′,

AlbS (reverse) 5′-GCTGTCATCTCTTGTGGGCTGT-3′, and

Albp (probe) 5′-CCTGTCATGCCCACACAAATCTCTCC-3′.

The samples (100 ng DNA) were analyzed in duplicate using the TaqMan masterMix^®^ (Roche Life Science, catalog no 04535286001) on a Rotor-Gene Q^TM^ (Qiagen) at 95 °C for 10 min, followed by 45 cycles at 95 °C for 30 s and 60 °C for 1 min. Quantification was measured compared to the standards obtained by the insertion of the amplified sequence (BMRF1 or albumin) into the pCR2.1 TA cloning vector^®^ (Invitrogen, Villebon-sur-Yvette, France). After purification of the two constructs by Qiagen Plasmid Maxi kit^®^ (Qiagen), the copy number was calculated after concentration determination by OD_260_ measurement. Then, serial 10-fold dilutions were made to prepare the standard curves. Knowing that there are two copies of the *albumin* gene per cell, the albumin copy number obtained for each sample was used to relatively determine the number of viruses per cell.

## 3. Results

### 3.1. Analysis of Whole EBV Genome Sequences

The complete analysis concerned 36 patient samples (34 patients with lymphoproliferative disease, all being selected because of EBER RNA detection in their biopsies, one control with a reactive adenopathy without any EBV pathology, and one control with a primary infection) and seven cell lines. The patient description is shown in Table 1. To obtain complete EBV genome sequences, a target capture method was used because viral loads may be low and because the EBV genome shares several homologies with the human genome. For 10 samples, the viral load was <100,000 copies/μg DNA, and a very low depth was obtained by MiSeq sequencing. Therefore, these samples were sequenced using NextSeq sequencing. The use of two different approaches to analyze the results, namely reference mapping and de novo assembly, which are complementary, allowed us to obtain more complete and accurate whole EBV genomes. The overall results are reported in Table 2. The mean read number per sample was 10,889,606, with a mean depth of 4136. An average of 96% of the reads mapped to the EBV genome, and all the sequences were analyzed after removing the low-percentage reads mapping to the hg19 human genome. Using the de novo assembly, the genomes obtained on the MiSeq sequencer were assembled into contigs created by Viramp and aligned to the reference genome to generate scaffolds. Seventeen whole genome sequences, including eight AITL were determined and deposited in the NCBI GenBank (N° MH837512 to MH837528). The individual sample accession numbers are listed in Table 3. The coverage profile was similar for all samples and greater than 92%. The mean percent GC content was 58.99%. The Fastq reads obtained for the totality of the samples have been deposited in NCBI GeneBank (BioProject ID PRJNA505149).

### 3.2. Some AITL Strains Exhibited Similar Distribution Patterns of Mutations

The whole genome sequences obtained for the patients were mapped to the reference genomes NC_007605.1 for EBV-1 and NC_009334.1 for EBV-2. Given that the latent *EBNA-2* and *-3* genes are the most divergent genes between EBV-1 and -2, we used the ratio “number of reads mapping to EBV-1 *EBNA-2*, *-3* genes/number of reads mapping to EBV-2 *EBNA-2*, *-3* genes” to determine the EBV type of each sample (Table 2). All but one patient sample (AIL18, which belonged to a native North African woman) harbored EBV-1. The raw reads were then aligned to the EBV-1 reference genome, and genetic variations were detected using the Varscan tool (Figure 1 and Appendix A). T Figure 2 illustrates the relative genetic variations among the genomes for the different strains compared to the reference. For the AITL patients, although no consensual sequence of the EBV genome was found, it is noticeable that AIL1/6/7/8/10/11/12/13/14/15/17 patients had very close strains. The same observation was true for the AIL3/4/5/9 patients. It is also noteworthy that two strains were particularly mutated: AIL2 and AIL16.

### 3.3. AITL Biopsies Revealed EBV in a Clonal Form

To determine whether the virus was clonal in biopsies, we calculated the heterogeneity, i.e., the number of heterogeneous positions, for each sample (Appendix A). Because the sequence depth was high using deep sequencing, we considered a position as heterogeneous if the variant frequency was between 20 and 94%, as proposed by Kwok et al. [49]. A low level of heterogeneity supports the monoclonal origin of a strain, whereas high heterogeneity is due to the presence of various strains. Here, to determine clonality, we chose a cut-off of 0.2% heterogeneity, which is more stringent than the 0.5% cut-off chosen by Kwok et al. In this condition, Control1 and PI1 presented heterogeneous EBV strains, and the cell lines harbored monoclonal strains (Figure 3). Interestingly, all the AITL samples tested contained EBV in clonal form. Only four patient biopsies (PTTL1, PTBL3, PTBL4, and LPL1) showed heterogeneity in favor of a non-clonal virus. We then correlated these findings to the B and/or T clonality for each AITL sample and found that all but one sample (AIL3) had a T clonality (94.4%), while B clonality was found for five samples (27.7%) (Figure 3).

### 3.4. EBV Was Almost Always Latent in AITL Biopsies

In order to establish whether EBV was in a latent or a replicative state in the biopsies studied, we determined the viral load in each sample by quantitative PCR (qPCR), targeting the *BMRF1* gene. The human *albumin* gene was also quantified in each sample using the same protocol in order to calculate the number of viruses per cell. Based on the publication of Hsieh et al. [50], a virus can be considered as latent if there are less than 20 copies per cell; a higher intracellular viral load is indicative of active replication. According to this calculation, only one AITL biopsy (AIL5) exhibited viral replication, while the others contained latent EBV (Table 4). Active replication was also found in other lymphoma samples (CTCL1, NK/TL2, PTBL1, DLBCL3, and ARL2).

### 3.5. Phylogenic Analysis Confirmed the Existence of at Least Two Different Groups of EBV among AITL Biopsies

The phylogenic study was conducted for all the sequenced strains, based on the multiple nucleotide sequence alignment of whole genomes (Figure 4). Unsurprisingly, it clearly individualized AIL18, the only type 2 EBV of this series. Furthermore, the AIL2 and AIL16 strains were clearly different from the vast majority of the AITL strains, which were categorized into two different groups: strain 1 and strain 2. Strain 1 was found in patients AIL1/6/7/8/10/12/13/14/15/17 and strain 2 corresponded to AIL3/4/5 and -9 patients, whose proximity to each other and to CTCL1 was noticeable. AIL3/4/5/9 patients had an average survival of 2.4 years after their diagnosis, while it was 7.1 years for the AIL1/6/7/8/10/11/12/13/14/15/17 patients; this difference was significant, as demonstrated by the Kaplan-Meier curve (Table 1 and Figure 5). Overall, phylogenetic analysis showed that there was no EBV strain specific to AITL. Phylogenetic analysis was also conducted on each of the latency genes. Interestingly, the two groups already described were clearly individualized for the *EBNA-3A*, *EBNA-3B*, *EBNA-3C*, and *LMP-2* genes (Figure 6). Appendix A shows the phylogenetic trees obtained for the EBNA-1, EBNA-2, and EBNA-LP genes.

### 3.6. Mutations Occurred Mostly in Latency Genes and Secondarily in Tegument Genes

The means of the single nucleotide variations (SNVs) and the insertions and deletions (INDELs) were not higher in the AITL group compared to other lymphomas (respectively, 265 and 402 for SNVs and 5 and 7 for INDELs). The analysis of non-synonymous mutations according to nine main gene categories (namely latency, replication, membrane glycoprotein, tegument, capsid, transcription, metabolism, packaging, and unknown function) showed that the majority of changes were located in latency genes, though a large number of mutations also occurred in tegument genes (Figure 7). Conversely, capsid and transcription genes had the lowest number of variations. For each gene, we calculated the average number of non-synonymous mutations for the “AITL” group versus the “other lymphoma” group. The results showed a significant difference between these two groups and are reported in Table 5. In addition to the latency and tegument proteins, it can be noted that some replication proteins were more mutated in the AITL group, such as the BKRF3 protein for example. We also looked for variations in proteins implicated in the switch from latency to the lytic cycle, mainly Rta, a product of the *BRLF1* gene, and Zta or ZEBRA, *BZLF1* encoded protein. The patients AIL2/3/4/5/9/16 harbored the BRLF1 mutation S542N and, except for AIL2, the BZLF1 mutation A206S (Figure 8). These two mutations were positioned in CD8 epitope sites, while the other patients harbored BRLF1 A290D, V479I, and P486S. It is noteworthy that AIL2 exhibited nine BZLF1 mutations (not represented in Figure 8).

### 3.7. EBNA-1, EBNA-2, and Mainly EBNA-LP Genes Were the Most Mutated among Latency Genes

The majority of strains originating from the AITL biopsies contained four *EBNA-1* mutations (E16Q, G18E, E24D, and G27S), located in the Gly-Arg domain (aa, amino acids, 8–67) and implicated in the EBNA-1-dependent DNA replication and partitioning of the EBV episomes in dividing cells (Figure 9). The region implicated in the other latency gene expression activation (aa 61–89) carried the mutation T85A present in almost all the strains derived from AITL and the mutations V70A and Q74P. Furthermore, the mutation T585P, which occurs in the dimerization domain of the protein, and which is located in important CD4 and CD8 recognition epitopes, was also present in all but one AITL strain. All AITL strains, except AIL2 and AIL11, presented a threonine at the signature codon 487 and therefore belonged to the P-thrV subtype, as described by Gutierrez at al. [51].

Among the three categories of *EBNA-2* domains critical for its transcription regulation function, two are particularly mutated among the AITL strains: the self-association domain 3 (SAD3) and one of the nuclear localization signals (NLS). Indeed, SAD3 (aa 148–214), especially, carried the mutations R163G, Q185R, M196I, T204S, and the conservative duplication L211, which were present in a large number of AITL biopsies. Similarly, the highly represented mutations, H316N on one hand and E476G and P478S, as well as S485P or the S485 frameshift, on the other hand, were located, respectively, on NLS1 (aa 284–341) and NLS2 (aa 471–487). Some of these mutations affected the epitopes recognized mainly by CD8 cells.

EBNA-LP contained a variable number of 66 amino acid repeats corresponding to the W1 and W2 exons of IR1, followed by a C-terminal non-repetitive domain encoded by the exons Y1 and Y2. Conserved regions were determined in the C extremity (CR1 to CR3, implicated in EBNA-2 binding) and in the N-terminal region (CR4, CR5). Although one must be prudent with regard to results obtained for repeats with fragmentation sequencing, the sequences obtained here showed significantly more substitutions in AITL EBNA-LPs compared to the others. The mutations observed for the AITL biopsies (H88N/Q/R, V94E and V101I) were all localized at exon Y2 and concerned the majority of strains.

It is noticeable that for these three proteins the AIL2 strain had an identical profile to the PI or control1 strains without any of the described mutations.

### 3.8. Two Tegument Genes, BNRF1 and BBRF2, Were Especially Mutated in AITL Biopsies

The major mutations concerning BNRF1 and BBRF2 are reported in Figure 8. The BBRF2 protein forms a hetero-complex with the tegument protein BSRF1, which mediates the viral envelopment [52]. For this protein, the AITL patients showed statistically more mutations than the other patients; A176S is particularly common in AILs.

The major tegument protein BNRF1 can bind to DAXX (death-domain associated protein-6) histone chaperone H3.3 and H4 to form a stable quaternary complex. BNRF1 also carries a PurM-like domain (aa 610–976) and a GATase domain (aa 1037–1318). Detected BNRF1 mutations were unevenly distributed according to the strains. For example, the two mutations, P580T and S587R, which occurred on the DAXX interaction domain (DID), aa 360–600, concerned seven AITL biopsies. Similarly, among the 3 mutations (A762V, N797S, and S861C) which affect the purM-like domain of the protein, 2 were found for 12 AITL biopsies. AIL3/4/5/9 exhibited a mutation profile different from the other strains. No mutations concerned the CD4 or CD8 epitopes.

## 4. Discussion

Although AITL is an uncommon, aggressive disease, it is one of the more frequent subtypes of PTCL in the western world. EBV is found in as high as 95% of cases [53], which highlights the close relationship between EBV and AITL. To date, however, whether EBV infection plays a role in AITL pathogenesis remains unclear, and diverse assumptions exist. For this reason, we decided to study the complete sequence of the EBV genome in AITL patients, compared to patients with other lymphomas.

Since EBV was discovered and characterized in BL [54], a B-cell lymphoma, it has been considered to be involved in other B-cell neoplasms. In these proliferations, EBV-encoded latent proteins have been shown to directly promote immortalization and proliferation through NF-kB pathway stimulation and increased anti-apoptotic gene expression. Concerning T cell lymphomas, and especially AITL, the role of EBV is probably much more complex and diverse. It has been shown that EBV is mainly present in AITL in B cells [32,55,56]; so, it is assumed that it plays an indirect role in the infected tumor cells through modification of the tumor microenvironment. In these conditions, it is essential to know whether the virus is latent or replicative as the proteins involved and the mechanisms of action are different. Unfortunately, for the biopsies we examined, there was not enough material to be able to determine which cells carried the virus. For these samples, with a particular interest in AITL biopsies, we explored the viral latent or replicative state, and we found that all but one AITL sample harbored only the latent virus (Table 4). To our knowledge, very few papers have examined whether the virus was latent in AITL. Unlike our results, Smith et al., arguing that EBV-positive cell numbers were greater than expected in reactive tissue and that infected cell nuclei were larger than average, reported an active replicative state of EBV [22]. Our method for determining whether the virus was latent [50] seems more accurate and is corroborated by the fact that small EBER RNAs were found in all the AITL cases. Interestingly, only one AITL patient (AIL5) had a high viral load, with an estimated copy number of 12,074 per cell, which is consistent with a replicative state of the virus. In this case, given the presence of EBERs, it can be assumed that some cells carry latent virus, while it is replicative in others. In addition to this work, we recently published a study of viral transcriptomes for seven of these patients (AIL2/3/7/11/14/15/16) and others, showing that the virus was in latency II and, more specifically, in latency IIc [57]. These results call into question the hypothesis that the virus could act in disease progression through cytokine and chemokine modulation. Indeed, in this hypothesis, the immunodeficient state resulting from the disease would lead to viral reactivation, thereby playing a role in the development of the tumor microenvironment [53]. It is more likely that EBV persists in a life-long latent state in infected cells and that this presence, in conjunction with other carcinogens, may promote the evolution to cancer [58].

For each sample, we then calculated the heterogeneity of the obtained sequence, namely the percentage of heterogeneous sites in relation to the genome size [59]. The variability threshold that we chose was 0.2%. This threshold, which is clearly below the threshold chosen by Kwok et al., was validated by the results obtained for the cell lines tested, which were considered as monoclonal controls, and for the primary infection sample and the inflammatory reactive biopsy, which were considered as polyclonal controls (Figure 3). The overall heterogeneity for AITL was between 0.015% (for AIL6 and AIL13) and 0.171% (for AIL2). The 18 AITL samples tested showed less than 0.2% variability, indicating that for these samples the heterogeneity was too low to be due to infection with various strains. As mentioned by others [49], rare mutations may spontaneously arise in viral genomes during clonal expansion. The low number of heterogeneous positions observed is consistent with this hypothesis, and the monoclonal origin of EBV can be attributed to it. This result, found for all AITL samples, makes it possible to consider a role for EBV in this pathology, which would here be more than a simple passenger, contrary to what is suggested by others [53,60]. As expected for this pathology, T lymphocytes were clonal in all but one sample, while B clonality was found in five samples. Unfortunately, due to a lack of material, we were unable to visualize in which cells the virus was present.

Overall, we did not observe any particular strain in the AITL biopsies. After alignment against the EBV-1 and EBV-2 reference strains, the overall results showed that only one patient (AIL18), a native of North Africa, had a type 2 EBV. The low number of EBV-2 in this series is not surprising given that the majority of the patients originate from France and, on the other hand, that EBV-2 is probably less pathogenic than EBV-1. Comparative analysis of all the mutations obtained for the AITL patients and the others does not reveal a strain characteristic of AITL (Figure 2). This figure shows that the mutation profiles obtained for the AIL1/6/7/8/10/11/12/13/14/15/17 strains are very close. Likewise, the mutation patterns of the AIL3/4/5/9 strains are quite similar; this is particularly visible in the EBNA-3A and -3B regions. It can also be noted that the strains of AIL16, and especially of AIL2, are particularly mutated. The phylogenic analysis carried out on entire sequences supports these different observations (Figure 4). Although many other factors are involved in disease progression, it is interesting to note that the AIL3/4/5/9/16 patients, whose viral strains are close, had an average survival of 2.1 years after their diagnosis, while it was 7.1 years for the AIL1/6/7/8/10/11/12/13/14/15/17 patients (Table 1). As demonstrated by the phylogeny and regarding the latency genes, the main differences between these two groups of strains relate to *EBNA-3A*, *EBNA-3B*, *EBNA-3C*, and *LMP-2* (Figure 6). For the AIL2/3/4/5/9/16 strains, we showed that there was an S542N mutation on the *BRLF1* gene encoding the Rta viral transcription factor. This mutation is positioned on the transactivation domain of the protein. This has also been described by Farrell P.J. (unpublished data) in T-cell disease in Paris. Similarly, the AIL3/4/5/9/16 strains carry the A206S mutation on the *BZLF1* gene, a mutation already reported by other authors [61,62], located on the dimerization domain of the viral transcription factor Zta. These two mutations (*BRLF1* S542N and *BZLF1* A206S) affecting important domains can cause a change in viral behavior. Note that the other strains featured the mutations V479I, P486S on the activation accessory domain and A290D on the DNA-binding domain of *BRLF1*. These mutations have already been reported [62,63].

Our initial idea that there may be a viral strain specific to AITL led us to analyze the main mutations found on viral genomes. Analysis of genome variation locations showed that the coding regions constituted more than 70% of the mutations, while the EBER and microRNA regions were conserved. Consistent with previous reports [8,59,64], we observed a higher frequency of non-synonymous SNPs in latent genes (Figure 7), followed by the genes encoding tegument proteins. Although the mutation mean was not higher for AITL than for the other lymphomas, the calculation, for each gene, of the number of non-synonymous mutations for the AITL group versus that number for the other lymphomas, showed a significant difference for some genes, especially the *EBNA-1*, *EBNA-2*, and *EBNA-LP* latency genes and the tegument *BBRF-2* and *BNRF-1* genes (Table 5).

EBNA-1 is very interesting to consider because it is the only EBV antigen consistently expressed in EBV-associated malignancies [1], expressed in all forms of latency, and because of its major function in viral episome maintenance in latently infected cells. In this regard, it is intriguing to see that most AITL strains (15/17) carry four substitutions on the Gly-Arg domain aa 8–67, which is mainly involved in the replication and partitioning of the EBV episomes in the two dividing cells, which are thus able to modify these functions (Figure 9). Apart from these mutations, the T85A substitution, already described by Borozan et al. [62] and found in all our AITLs except AIL2, could play an important role in proliferation because it is positioned in a region involved in the transcriptional activation of other latency genes. The V70A and Q74P mutations present in 10 AITL, which are not described to our knowledge, may contribute to this effect, being positioned in the same region. Finally, the T585P mutation was found in all AITL except AIL2. Recently, the DNA-binding domain of EBNA-1 has been described to be in the form of an oligomeric hexameric ring, the oligomeric interface pivoting around residue T585 [65]. T585 is a position subject to substitutions. Mutations occurring on this residue had effects on EBNA-1-dependent DNA replication and episome maintenance [66]. Polymorphism at this position is often found in NPC tumors and Burkitt’s lymphoma [65]. We compared the sequences obtained for our patients to the sequences obtained for the four LCLs (DPL, KREB2, CoAN, and MLEB2) that we established from non-hematological subjects [57]. Among the different interesting mutations found in our present patients and concerning different genes, only *EBNA-1* T85A and *EBNA-1* T585P were found in LCL. These mutations seem to be widespread in the French population. It is noteworthy that the changes observed in the EBNA-1 sequences fell principally within the known T-cell epitopes: all mutations but E24D and G27S are located in CD8 epitopes and V70A, Q74P, T85A, and T585P are positioned in CD4 epitopes. These results imply that immune pressure could be in part responsible for these changes. Based on the amino acid present at EBNA-1 position 487, EBV has been classified into five subtypes: P-ala (wild type B95-8 subtype), P-thr, V-leu, V-val, and V-pro [51,67]. Considering these five subtypes, it is noteworthy that 15 among the 17 AITL EBV-1 positive patients harbored a P-thr subtype, which was also the most prevalent in the other samples. V-val is reported to be dominant in Asian regions, while the P-thr subtype is most commonly observed in the peripheral blood of European and Australian subjects, as well as in African tumors [63].

The latent protein EBNA-2 acts mainly as a transcription factor and therefore comprises the transactivation domains (TAD), auto-association domains (SAD), and nuclear location signals (NLS) important for its function. The main mutations affecting EBNA-2 in the AITL group impact the SAD3 domain or the NLS domains and are thus able to modify its regulatory function. Many of these mutations, particularly the E476G mutation, present among others in the E2A subtype [68], have been described in the French as well as English or Eurasian, American [62], or Asian [69] populations. To our knowledge, only the H316N mutation located on the NLS1 domain and the R163G positioned on the SAD3 domain have not been previously described, although position 163 was found to be very frequently mutated (R163M substitution) by Wang et al. [68]. These two mutations are highly represented in AIL.

EBNA-LP is a latent protein, acting principally as an EBNA-2 coactivator and therefore has an important role in B-cell immortalization. It is notable that this was the most significantly mutated protein in the AITL group compared to the other patients. The main substitutions observed for our patients (H88N/Q/R, T93P, V94E, and V101I) were grouped and located at the C-terminal extremity, between CR4 and CR5. Most published viral sequences do not carry these mutations, apart from V94E and V101I, which were described by Ba Abdullah et al. to characterize subgroup C [70].

Two tegument proteins were particularly more mutated in AITL patients than in other patients: BBRF2 and BNRF1 (Figure 8). Recently, it was shown that BBRF2 can associate with BSRF1, another tegument protein, and the heterocomplex formed has a role in the binding of EBV nucleocapsids to the Golgi membrane during secondary envelopment. Therefore, BBRF2 appears to play an important role in viral infectivity [71]. The main mutation found (A176S) is highly represented in AIL compared to other pathologies. However, it has already been described, including in healthy subjects [4,61].

The major tegument protein BNRF1, through its binding to the antiviral DAXX histone chaperone, H3.3 and H4, is implicated in the establishment of latency and cell immortalization [72]. The stable quaternary complex formed is responsible for its localization to the PML-nuclear bodies involved in the antiviral intrinsic resistance and the transcriptional repression of host cells [73]. The mutations that we have highlighted in AITL only concerned a subgroup of patients. Most of these mutations have been previously described [4,62]. However, some of them could alter the behavior of the protein, such as those located on DID, namely P580T and S587R, or those affecting the purM-like domain of the protein (A762V, N797S, and S861C).

Interestingly, BKRF3, a viral uracil-DNA glycosylase, participates in the repair of viral DNA and prevents its mutagenesis. Obviously, for this protein, the mutations that severely affect viral replication cannot be retained. In this study, E128D substitution was present for all AITL patients except AIL2. The presence of this mutation could modify the virus–cell relationships.

In the end, it is obvious that there is probably no specific viral mutation for AIL. However, on the other hand, it is possible that a combination of mutations affecting several genes, even if they have already been listed, could modify the behavior of the virus. With this in mind, it would be interesting to study how the virus found in AIL3/4/5/9 behaves within the cell.

## 5. Conclusions

In summary, the whole viral sequences obtained for the 18 AITL patients and compared to other patients identified a poorly represented strain within the AITLs, but which seemed to be associated with poor outcome. Furthermore, we demonstrated that the virus was clonal and latent in all the AITL biopsies analyzed. These various elements suggest that the virus is involved in this pathology.

## Figures and Tables

**Figure 1 cancers-14-02899-f001:**
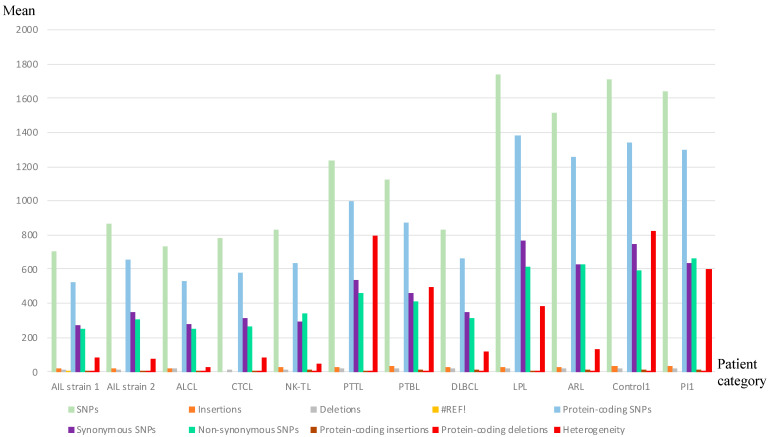
Mean genetic variations and heterogeneity determination for all categories of patients.

**Figure 2 cancers-14-02899-f002:**
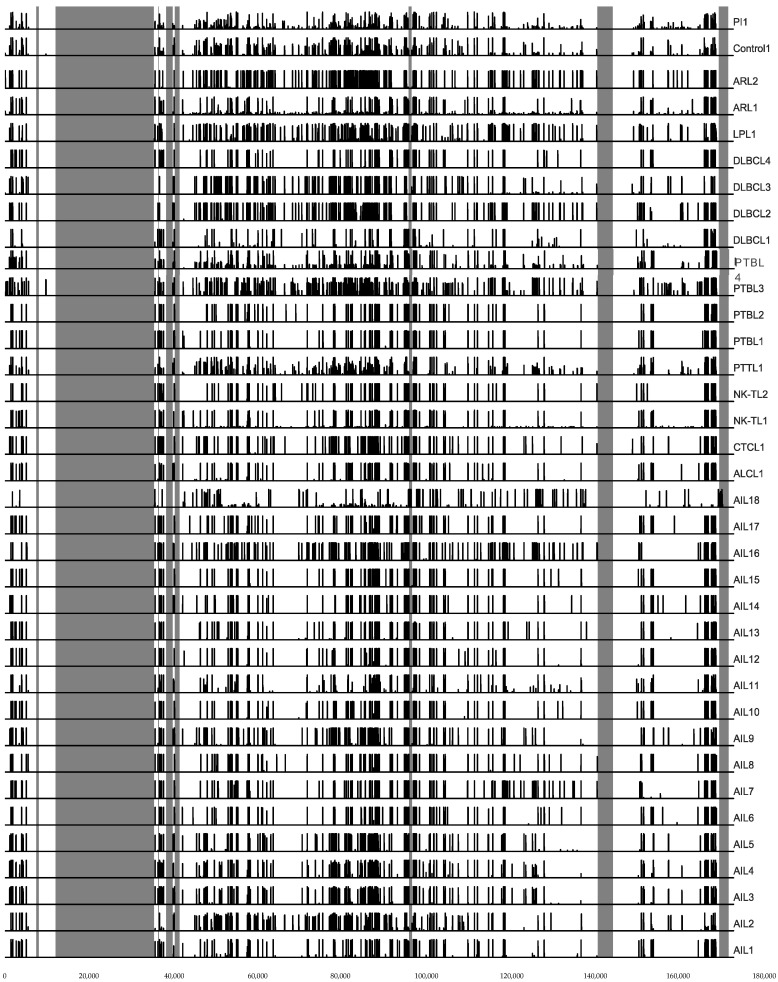
Genetic variations among the genome for the different viral strains compared to the reference. Regions shaded in gray correspond to internal and terminal repeats.

**Figure 3 cancers-14-02899-f003:**
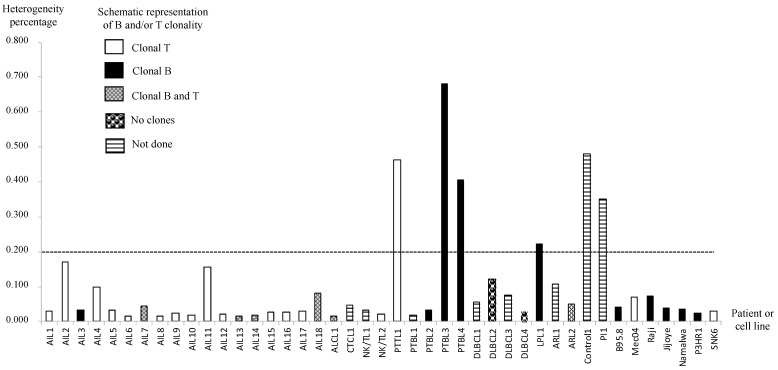
EBV clonality in AITL and other lymphoma samples. Viral clonality was assessed based on the work of Kwok et al. [33]. B and/or T clonality is also represented for each sample.

**Figure 4 cancers-14-02899-f004:**
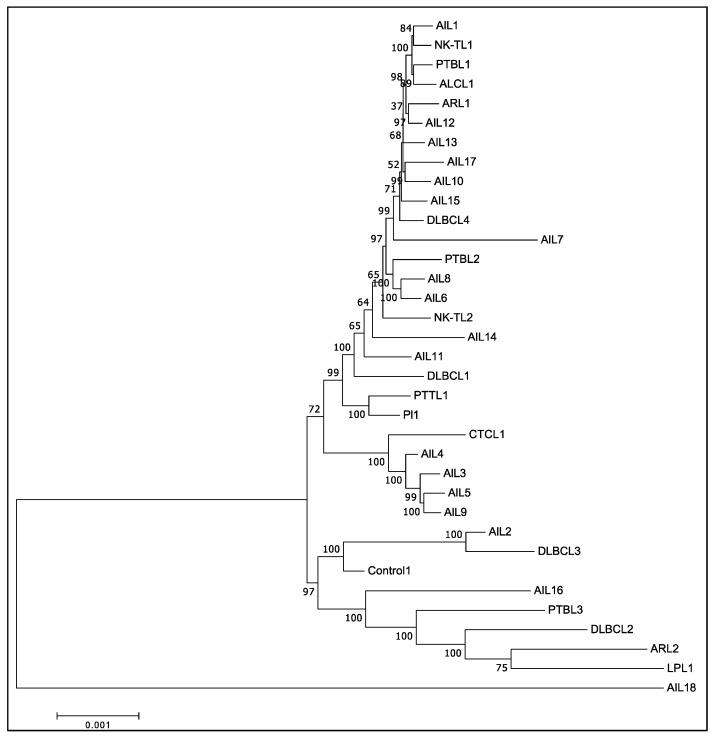
Phylogenetic tree obtained after whole genome nucleotide sequence alignment of the different strains.

**Figure 5 cancers-14-02899-f005:**
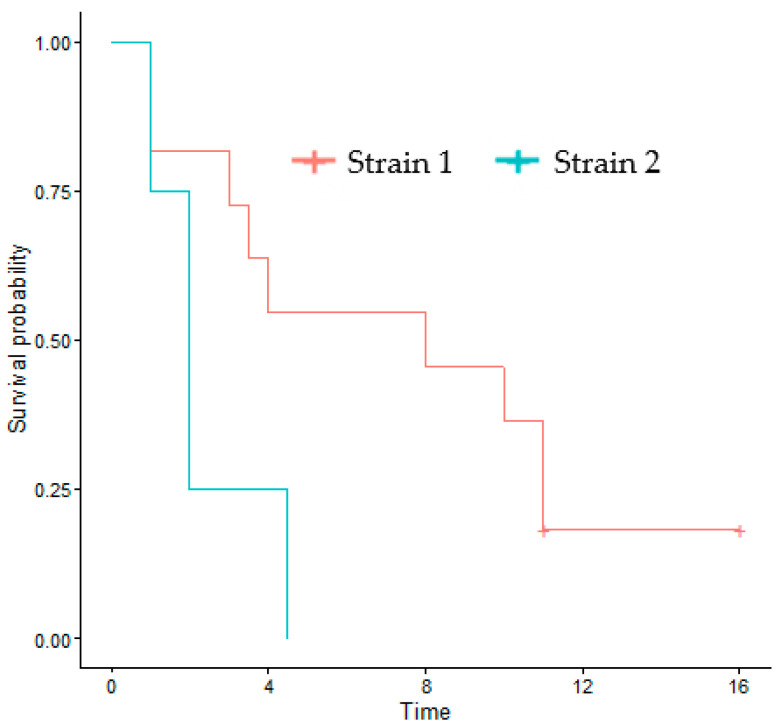
Kaplan-Meier curve showing significant differences in the survival of AITL patients carrying strain 1 compared to those carrying strain 2 (*p* = 0.048).

**Figure 6 cancers-14-02899-f006:**
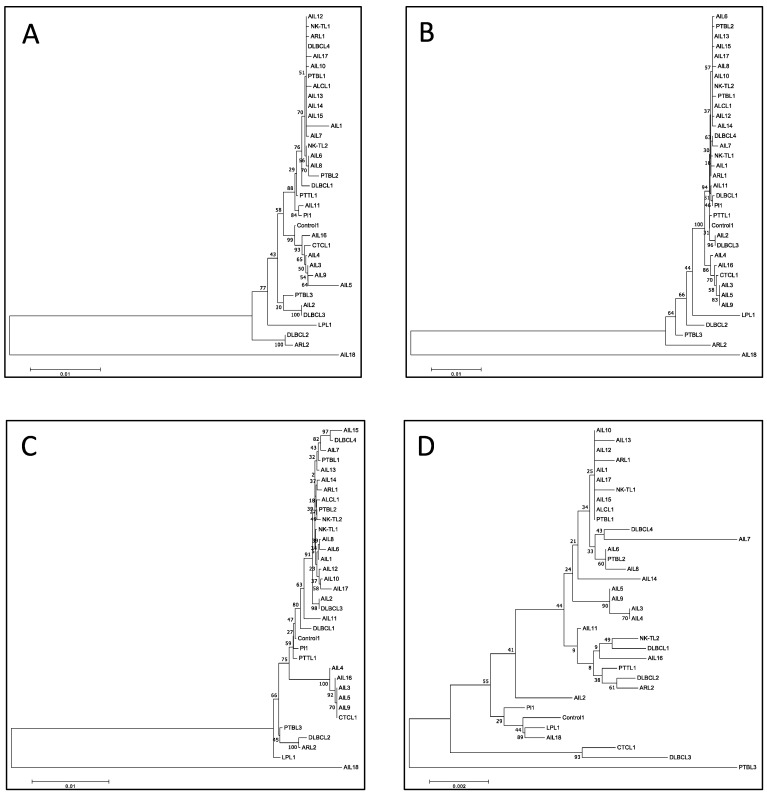
Phylogenetic tree obtained after nucleotide sequence alignment of the different strains. (**A**) EBNA-3A, (**B**) EBNA-3B, (**C**) EBNA-3C, (**D**) LMP-2.

**Figure 7 cancers-14-02899-f007:**
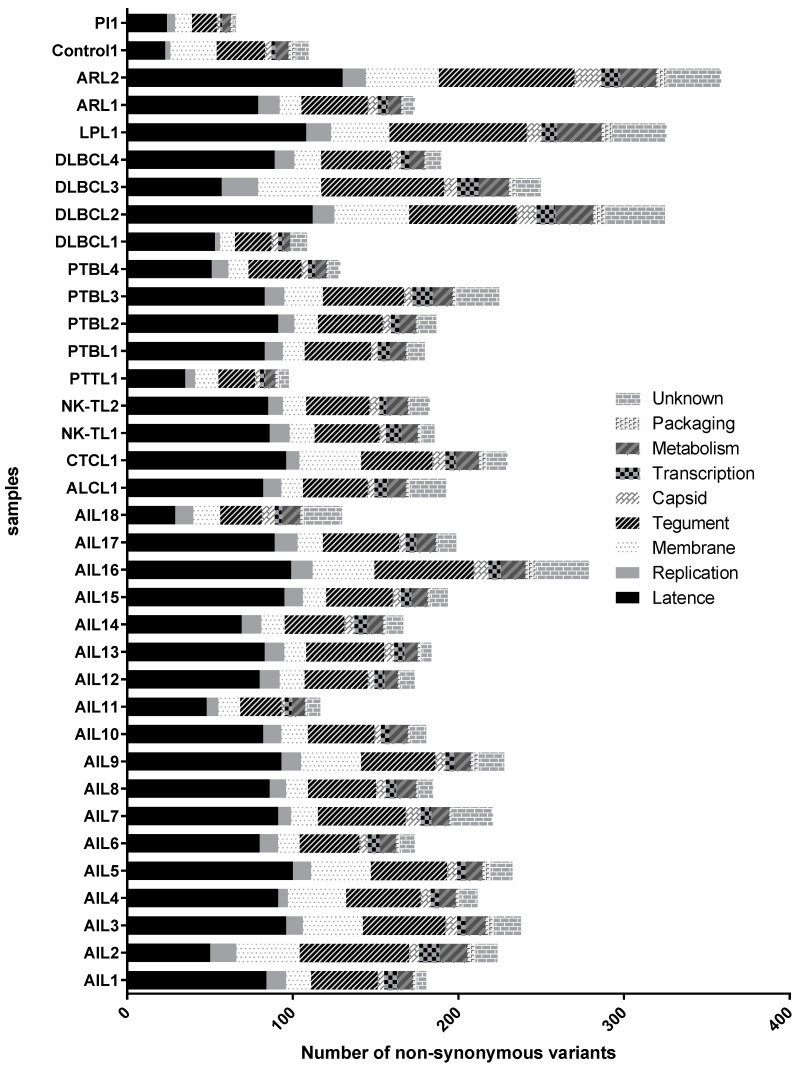
EBV non-synonymous mutation analysis according to nine main gene categories: latency, replication, membrane glycoprotein, tegument, capsid, transcription, metabolism, packaging, and unknown function.

**Figure 8 cancers-14-02899-f008:**
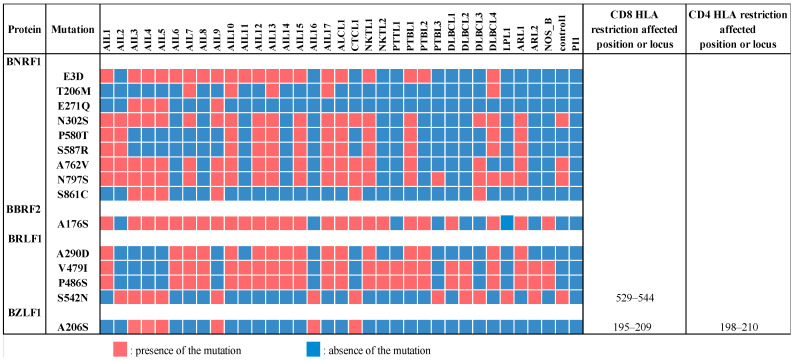
Main mutations observed among the tegument proteins BNRF1 and BBRF2, and the transcriptional activators BRLF1 and BZLF1. The CD4 and CD8 HLA restriction positions affected by the mutations are reported. For each protein, some mutations (not mentioned here) were present in all samples, including Control1 and PI1 (BNRF1 E36Q, C736F, and F1110S, BRLF1 A377E). Such variations were considered as geographically restricted. Mutations of AIL18, which is an EBV-2, are not reported.

**Figure 9 cancers-14-02899-f009:**
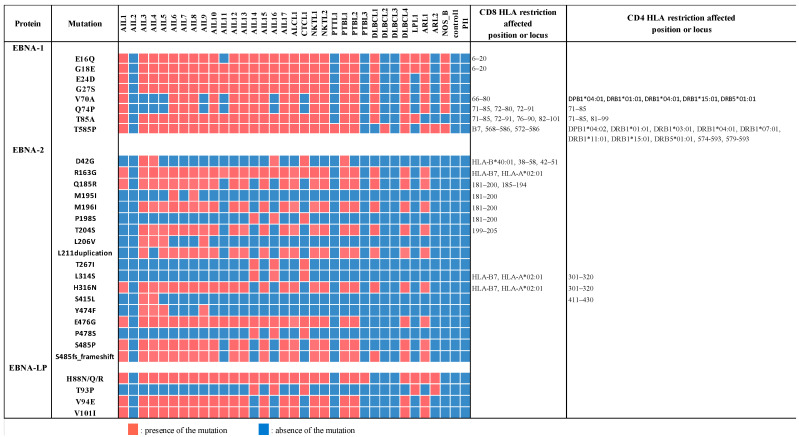
Main mutations observed among the latency proteins EBNA-1, EBNA-2, and EBNA-LP. CD8- and CD4-HLA restriction-affected position or locus are mentioned. Mutations of AIL18, which is an EBV-2, are not reported.

**Table 1 cancers-14-02899-t001:** Characteristics of the EBV-Associated Lymphoma Patients Included in this Study. Patients AIL10, AIL13, AIL18, NK/TL1, NK/TL2, PTBL1, PTBL3, PTBL4, LPL1 and ARL2 Are Still Alive Today.

Patient	Sex	Age at Diagnosis	Pathology According to WHO Criteria (2016)	Survival after Diagnosis (Years)
AIL1	F	73	Angioimmunoblastic T-cell lymphoma	1
AIL2	M	62	Angioimmunoblastic T-cell lymphoma	10
AIL3	M	80	Angioimmunoblastic T-cell lymphoma	2
AIL4	M	55	Angioimmunoblastic T-cell lymphoma	2
AIL5	F	66	Angioimmunoblastic T-cell lymphoma	4.5
AIL6	F	84	Angioimmunoblastic T-cell lymphoma	1
AIL7	F	59	Angioimmunoblastic T-cell lymphoma	4
AIL8	F	52	Angioimmunoblastic T-cell lymphoma	11
AIL9	F	76	Angioimmunoblastic T-cell lymphoma	1
AIL10	F	53	Angioimmunoblastic T-cell lymphoma	>16
AIL11	M	59	Angioimmunoblastic T-cell lymphoma	10
AIL12	F	56	Angioimmunoblastic T-cell lymphoma	11
AIL13	F	78	Angioimmunoblastic T-cell lymphoma	>10
AIL14	M	62	Angioimmunoblastic T-cell lymphoma	3
AIL15	M	67	Angioimmunoblastic T-cell lymphoma	3.5
AIL16	M	50	Angioimmunoblastic T-cell lymphoma	1
AIL17	F	84	Angioimmunoblastic T-cell lymphoma	8
AIL18	F	67	Angioimmunoblastic T-cell lymphoma	>10
ALCL1	F	73	Anaplastic large T cell lymphoma	3.5
CTCL1	F	73	Cutaneous T cell lymphoma	1
NK/TL1	F	51	NK/T cell lymphoma	>21
NK/TL2	M	64	NK/T cell lymphoma	>9
PTTL1	M	63	Post-transplant T lymphoma	1.5
PTBL1	M	28	Post-transplant B lymphoma	>5
PTBL2	F	63	Post-transplant B lymphoma	<1
PTBL3	F	65	Post-transplant B lymphoma	>10
PTBL4	F	52	Post-transplant B lymphoma	>15
DLBCL1	F	74	Diffuse large B-cell lymphoma	2.5
DLBCL2	M	77	Diffuse large B-cell lymphoma	<1
DLBCL3	F	59	Diffuse large B-cell lymphoma	4.5
DLBCL4	F	31	Diffuse large B-cell lymphoma	10
LPL1	F	18	Lymphoplasmacytic lymphoma	>7
ARL1	F	89	Age related lymphoma	<1
ARL2	F	76	Age related lymphoma	>8
Ct1	F	73	Reactive adenopathy	
PI	M	13	EBV primary infection	

**Table 2 cancers-14-02899-t002:** Overall Results Obtained for Whole EBV Genome Sequencing.

Sample	Number of Reads	Mapping to EBV (%)	Mapping to hg19 (%)	Mean Depth	Genome Coverage 10× (%)	*EBNA-2,3* EBV-1/*EBNA-2,3* EBV-2 Read Numbers
AIL1	73,730	97.86	6.76	68	97.1	6.85
AIL2	105,196	97.03	7.24	82	95.5	6.15
AIL3	23,145,280	96.89	3.38	9,235	99.9	6.99
AIL4	59,822	98.18	12.11	46	96.8	7.07
AIL5	128,648	97.73	7.13	100	98.5	5.57
AIL6	13,342,518	98.28	22.04	4,426	99.8	8.24
AIL7	922,026	98.73	5.4	779	99.8	6.18
AIL8	9,076,710	97.99	29.91	2,702	99.8	8.38
AIL9	34,991,168	98.67	11.49	13,180	99.9	8.59
AIL10	27,435,252	98.99	9.46	10,600	99.9	7.80
AIL11	2,827,228	94.83	50.35	582	98	12.48
AIL12	38,416,258	99.2	7.46	15,199	99.9	8.46
AIL13	74,846	96.91	13.01	58	96.8	6.03
AIL14	1,231,958	98.89	5.03	1,056	99.8	6.21
AIL15	1,047,872	98.84	5.28	875	99.7	6.41
AIL16	101,308	98.16	10.22	77	98.3	5.78
AIL17	53,864,512	99	7.72	21,194	99.9	9.21
AIL18	12,827,446	97.91	20.05	4,210	99.8	0.39
DLBCL1	2,529,424	94.2	53.57	485	93.5	8.52
DLBCL2	27,943,648	98.3	5.38	22,975	98.3	5.57
DLBCL3	20,223,716	98.06	5.55	16,076	98.2	6.28
DLBCL4	393,996	98.4	6.34	316	99.3	6.42
LPL1	102,642	96.7	10.26	75	97.7	3.57
PTBL1	4,050,570	98.34	5.02	3,184	99.8	6.26
PTBL2	2,588,074	98.48	4.64	2,139	99.8	6.45
PTBL3	828,392	97	6.4	630	99.8	14.44
PTLB4	624,062	98.42	5.07	513	99.7	6.08
ARL1	328,098	78.44	5.95	159	98.6	7.83
ARL2	1,772,596	98.26	5.47	1,435	99.5	2.21
PTTL1	1,777,550	75.57	39.28	437	99.4	7.05
ALCL1	1,021,930	98.38	5.83	116	99	6.40
CTCL1	18,578,118	98.63	5.32	15,039	96.1	5.60
NK/TL1	342,062	98.16	6.1	170	98.9	7.12
NK/TL2	6,742,818	98.43	4.74	5,689	99.9	6.52
Control1	841,766	77.41	7.86	495	99.4	6.24
PI1	13,1045,844	96.05	2.61	604	99.9	2.33
B95.8	17,718,578	98.6	7.33	15,283	98.5	6.13
Mec04	715,980	97.99	6.87	619	99.8	6.60
Raji	1,820,758	97.63	6.66	1,460	98.4	4.90
Jijoye	3,813,610	98.15	6.43	3,124	98.3	0.39
Namalwa	944,054	97.56	5.58	780	96.2	0.39
P3HR1	99,174	80.61	11.67	69	92	0.30
SNK6	1,733,840	98.05	7.04	1,487	96.7	0.41

**Table 3 cancers-14-02899-t003:** Characteristics of EBV Genome Sequences Created by Contig de novo Assembly and GenBank Accession Number for 17 Strains.

Sample	Contigs > 1000	Total Length of Contigs > 1000 bp	Contig Number	Total Length	Largest Contig	GC%	N50 Size	GenBank Accession Number
AIL1	9	142,548	11	143,842	52,398	57.81	36,930	MH837512
AIL2	19	144,946	33	154,536	32,793	58.14	9,443	MH837513
AIL5	24	153,365	41	164,471	22,826	58.8	11,438	MH837514
AIL7	11	172,444	12	173,126	50,944	59.45	43,411	MH837515
AIL13	29	152,897	47	164,966	35,094	58.74	8,301	MH837516
AIL14	4	169,984	7	171,612	80,073	59.53	73,612	MH837517
AIL15	5	170,653	5	170,653	81,106	59.5	71,452	MH837518
AIL16	13	147,983	21	153,913	55,163	57.89	44,505	MH837519
CTCL1	8	162,594	12	165,345	43,990	59.14	43,099	MH837521
NK/TL2	3	170,356	6	172,153	96,347	59.53	96,347	MH837524
PTBL1	5	171,454	6	172,258	71,791	59.59	50,232	MH837525
PTBL2	4	171,267	4	171,267	80,971	59.16	73,531	MH837526
PTBL3	9	169,947	13	172,617	63,345	59.38	28,613	MH837527
PTBL4	12	174,524	13	175,108	49,504	59.46	39,096	MH837528
DLBCL2	10	163,503	15	166,737	66,384	59.36	57,091	MH837522
DLBCL4	17	171,309	20	173,272	36,271	59.36	11,510	MH837523
ARL2	8	169,962	11	172,116	71,378	59.34	28,847	MH837520

**Table 4 cancers-14-02899-t004:** Latence/Reactivation State of EBV in Biopsy Samples.

Sample	EBV (Copies/μg DNA)	EBV (Copies/Cell)	Latence (L) or Reactivation (R)
AIL1	281,630	0.081	L
AIL2	413,060	0.082	L
AIL3	13,380	0.008	L
AIL4	150.324	0.348	L
AIL5	1,092,700	12,074.033	R
AIL6	31,130	0.002	L
AIL7	3,846,140	4.498	L
AIL8	60,490	0.003	L
AIL9	82,314	0.01	L
AIL10	156,480	0.012	L
AIL11	33,950	0.004	L
AIL12	59,830	0.005	L
AIL13	189,538	0.016	L
AIL14	2,713,100	0.185	L
AIL15	3,363,280	0.33	L
AIL16	372,724	7.499	L
AIL17	60,740	0.007	L
AIL18	40,080	0.006	L
ALCL1	141,168	0.065	L
CTCL1	299,914,110	413.674	R
NK/TL1	688,300	0.531	L
NK/TL2	22,046,280	27.731	R
PTTL1	74,270	0.117	L
PTBL1	18,916,830	34.084	R
PTBL2	17,616,830	16.698	L
PTBL3	5,774,190	6.451	L
PTLB4	7,021,700	9.002	L
DLBCL1	44,870	0.01	L
DLBCL2	105,109,230	12.012	L
DLBCL3	106,311,737	556.606	R
DLBCL4	122,317	0.15	L
LPL1	363,043	0.637	L
ARL1	110,690	17.295	L
ARL2	21,677,340	58.986	R

**Table 5 cancers-14-02899-t005:** Comparison of the Mean Number of Non-Synonymous Mutations/Gene Between the EBV “AITL” Strains and the “Other Lymphoma” Strains by the Mann-Whitney Test. Only Genes for Which *p* < 0.05 Were Reported.

Gene	Protein Function or Location	AIL	Other Lymphomas	*p*
*EBNA-LP*	Latency protein	4.76	3.06	0.0042417244
*BKRF3*	Replication protein	1.00	0.56	0.0049754187
*BBRF2*	Tegument protein	1.00	0.63	0.0108844758
*BORF2*	Replication protein	0.35	1.31	0.0133170812
*BBLF2*/*BBLF3*	Replication protein	2.82	1.88	0.0214000908
*EBNA-2*	Latency protein	9.35	6.31	0.0295629871
*BORF1*	Capsid protein	1.24	1.81	0.0329743486
*BNRF1*	Tegument protein	7.82	6.13	0.0393795760
*BMRF1*	Replication protein	0.12	0.44	0.0406785170
*EBNA-1*	Latency protein	15.76	13.44	0.0411169198
*BALF3*	DNA cleavage and packaging protein	0.18	0.75	0.0419918591
*BBLF4*	Replication protein	4.82	4.13	0.0427272346
*BFRF3*	Capsid protein	0.94	1.13	0.0447940279
*BGRF1*/*BDRF1*	DNA cleavage and packaging protein	0.35	0.06	0.0452271888

## Data Availability

The seventeen generated whole genome sequences were deposited in the NCBI GenBank (N° MH837512 to MH837528). The totality of the Fastq reads obtained have been deposited in NCBI GeneBank (BioProject ID PRJNA505149).

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
