# Peer review of "Epstein-Barr Virus (EBV) Is Mostly Latent and Clonal in Angioimmunoblastic T Cell Lymphoma (AITL)"

_cancers, 2022, doi:10.3390/cancers14122899_

Round 1

Reviewer 1 Report

Overall, this manuscript examines complete viral EBV genomes in 18 AITL patient samples by target-capture NGS. De novo assembly was used to reconstruct the resulting EBV genomes, examine the phylogenetic relatedness, assess mutations, and determine latency. Comparison to a small number of other EBV-associated T- and B-cell lymphomas was included. The study is significant for addressing an interesting aspect of AITL and EBV biology, including showing that predominantly two viral EBV strains were identified in these AITL cases. In addition, EBV was found to be clonal and latent in nearly all cases examined. However, the findings should be better presented.

General:

  1. Given that sample selection is important in such studies, was the pathology re-reviewed to confirm the initial diagnosis? Was a pathologist involved in the study and writing the manuscript? (needs further clarification in Method section 4.2, line 470). Was the pathology and clinical course re-reviewed for the patient with apparently replicative EBV? Supplementary figures with a representative image of AITL and the control 1 tissue and the corresponding EBER ISH would be helpful.

  1. The authors conclude that in the AITL cohort, two viral strains are present and they argue that one of these is associated with poor outcomes, however a statistical analysis such as a Kaplan-Meier curve is not shown to support this statement.

  1. Do the authors have any molecular somatic mutational data on the AITL and other lymphoma cases that are included in this study, which would support the diagnostic assessment? If available, these would be helpful to provide in Table 1 or in a supplementary table.

  1. The EBV reference contig (NC_007605) is included in the hg19. Was this taken into account for the de novo assembly when reads mapping to hg19 were removed?

  1. In Figures 3 and 4 and Table 6, there are subtle differences between AITL and a heterogenous group of “other” lymphomas. These should not be over-interpreted. However, it would be curious if a more homogenous and larger comparison group (like EBV+ DLBCL) would help tease out any differences. Would this be possible?

  1. Can the phylogenetic analysis shown in figure 4 be performed on the most mutated proteins described in section 2.7 {EBNA-1, EBNA-2 and Mainly EBNA-LP Genes Were the most Mutated Among Latency Genes)?

Specific comments:

The order of the manuscript sections is not correct based on the requirements of the journal. Please correct as follows: Introduction, Materials and Methods, Results, Discussion, Conclusions.

What did EBER ISH show from the tissued used as control 1?

Line 58: “According variations in the latent genes” can be edited to “Based on variations in the latent genes”

Line 61: Need reference for: “EBV-1 is also known to be more virulent than EBV-2.”

Line 120: correct term is “next generation sequencing (NGS)”

Table 1: LPL is spelled Lymphoplasmacytic lymphoma

Tables 2-5 contain green triangles that are distracting and should be removed.

Table 2: Other than GC content, were any other quality metrics applied to the sequencing data at the sample level (example minimum total reads)? Was a negative control used? The mean depth shows a lot of variability; what is this reflective of?

Table 3: Why only 17 EBV genomes were assembled out of this cohort?

Table 4 is confusing. Can it be moved to supplementary and instead the data can be summarized as a bar graph by grouping the two AITL strains and showing averages. Similarly, the other lymphomas can be grouped based on their classification.

Figure 1: Right-handed axis has text overlapping. Please fix.

Overall, the writing is unclear in places and lacks proper editing. Example: Line 443 delete “each time” between two complete sentences.

English phrasing is awkward in several instances:

Line 152: Awkward wording “sequences obtained for patients were mapped toward reference”; use “sequences obtained for patients were mapped to the reference”;

Line 173-175: Rewrite, awkward: “Using deep sequencing, a position was considered as heterogeneous if variant frequency was between 20 and 94%, as proposed by Kwok et al. [32], and thanks to the high sequence depth.”

Lines 318-321: Awkward phrasing in English, consider re-writing: “Although our method for determination of viral latency is only one approach to the true biology [33], it may be more accurate and corroborated by the fact that small EBER RNAs were found in all AITL cases.”

Figure legends 6 and 7 lack essential details for comprehension of the data: what the colors mean is not indicated.

Author Response

Dear Reviewer 1,

Please find attached the revised form of our manuscript ID: cancers- 1699745.

First of all, we would like to thank you for your very carefu reading and your very constructive comments which will improve our manuscript.

Please find enclosed your comments along with our responses.

General:

  1. Given that sample selection is important in such studies, was the pathology re-reviewed to confirm the initial diagnosis? Was a pathologist involved in the study and writing the manuscript? (needs further clarification in Method section 4.2, line 470). Was the pathology and clinical course re-reviewed for the patient with apparently replicative EBV? Supplementary figures with a representative image of AITL and the control 1 tissue and the corresponding EBER ISH would be helpful.

For pathological diagnosis, the slides were examined by two pathologists independently. The text (line 470) has been modified to make it clearer.

For patient AIL5 who has both latent and replicative EBV, the slides were re-read by a pathologist who said that there is too much variation within the AILs to be able to assign a particular character to the presence of a replicative EBV. Moreover, from a statistical point of view, one case is not sufficient to ensure any concordance between a given trait and the presence of a replicative EBV. Similarly, the clinic course for this patient was not informative. Alain Chaunavel was added in the authors and Dr Manuella Delage-Corre appears in the acknowledgments. A supplementary figure with control 1 and AITL EBER ISH tissues was added. 

  1. The authors conclude that in the AITL cohort, two viral strains are present and they argue that one of these is associated with poor outcomes, however a statistical analysis such as a Kaplan-Meier curve is not shown to support this statement.

Concerning the 2 main strains highlighted in AITL patients, a Kaplan-Meier curve was produced with calculation of significance. This has been added to the text.

  1. Do the authors have any molecular somatic mutational data on the AITL and other lymphoma cases that are included in this study, which would support the diagnostic assessment? If available, these would be helpful to provide in Table 1 or in a supplementary table.

Unfortunately, We have no information on somatic mutations for these patients

  1. The EBV reference contig (NC_007605) is included in the hg19. Was this taken into account for the de novo assembly when reads mapping to hg19 were removed?

Yes, this was taken into account.

  1. In Figures 3 and 4 and Table 6, there are subtle differences between AITL and a heterogenous group of “other” lymphomas. These should not be over-interpreted. However, it would be curious if a more homogenous and larger comparison group (like EBV+ DLBCL) would help tease out any differences. Would this be possible?

We totally agree with this comment. It is obvious that the sample size is too small and therefore the interpretation must be very careful. Unfortunately, this is due to the small size of our center and the many studies conducted by colleagues making it difficult to obtain well-documented samples large enough to carry out the work.

  1. Can the phylogenetic analysis shown in figure 4 be performed on the most mutated proteins described in section 2.7 {EBNA-1, EBNA-2 and Mainly EBNA-LP Genes Were the most Mutated Among Latency Genes)?

An additional figure has been added with the phylogenetic trees of EBNA-1, EBNA-2, and EBNA-LP.

Specific comments:

The order of the manuscript sections is not correct based on the requirements of the journal. Please correct as follows: Introduction, Materials and Methods, Results, Discussion, Conclusions.

The sections of the manuscript have been put in the required order

What did EBER ISH show from the tissue used as control 1?

Control 1 had a negative EBER staining as mentioned in supplementary figure 1.

Line 58: “According variations in the latent genes” can be edited to “Based on variations in the latent genes”

This was done

Line 61: Need reference for: “EBV-1 is also known to be more virulent than EBV-2.”

A reference was added

Line 120: correct term is “next generation sequencing (NGS)”

This was modified

Table 1: LPL is spelled Lymphoplasmacytic lymphoma

This was modified

Tables 2-5 contain green triangles that are distracting and should be removed.

The green triangles were removed

Table 2: Other than GC content, were any other quality metrics applied to the sequencing data at the sample level (example minimum total reads)? Was a negative control used?

The initial cohort included more patients than those mentioned in the article. Samples with mean depth <40 were not considered in further analysis (4 samples, namely AIL19, AIL20, PTCL-NOS1 and PTCL-NOS2). Otherwise, the samples showing a percentage of genome coverage below the accepted cut-off (85% for genome coverage 20X) were also excluded (more precisely the 4 previous samples plus DLBCL5 and DLBCL6).Considering the fact that spontaneous mutations may occur during strain culture, and considering the probability of some sequencing errors, the resequenced viral strains are highly similar to original sequences. Five genes, two latent genes (LMP1 and EBNA-3A) and three lytic genes (BMRF1, BORF1 and BFRF1, were also sequenced by Sanger method for the seven cell lines. Results obtained were perfectly similar showing exactly the same mutations for the BMRF1 gene from the Raji strain (mutations C67722T, C67746A and C67787T).

All these explanations have not been included to avoid complicating the text, but they can be added if necessary.

Of course, a negative control was included in each run.

The mean depth shows a lot of variability; what is this reflective of?

Some samples were old and/or carried little virus, which may explain the mean depth variability.

Table 3: Why only 17 EBV genomes were assembled out of this cohort?

For some samples the amount of virus was low and did not give results by MiSeq sequencing. They underwent fragmentation into small fragments and were sequenced by NextSeq. For these samples, the assembly results did not seem reliable enough to us and we only uploaded the FastQ files.

Table 4 is confusing. Can it be moved to supplementary and instead the data can be summarized as a bar graph by grouping the two AITL strains and showing averages. Similarly, the other lymphomas can be grouped based on their classification.

Table 4 has been placed in the supplementary tables and a figure has been introduced instead with the mean of the values ​​for each category

Figure 1: Right-handed axis has text overlapping. Please fix.

It has been corrected

Overall, the writing is unclear in places and lacks proper editing. Example: Line 443 delete “each time” between two complete sentences.

This was done

English phrasing is awkward in several instances:

Line 152: Awkward wording “sequences obtained for patients were mapped toward reference”; use “sequences obtained for patients were mapped to the reference”;

This was modified

Line 173-175: Rewrite, awkward: “Using deep sequencing, a position was considered as heterogeneous if variant frequency was between 20 and 94%, as proposed by Kwok et al. [32], and thanks to the high sequence depth.”

This was re-written

Lines 318-321: Awkward phrasing in English, consider re-writing: “Although our method for determination of viral latency is only one approach to the true biology [33], it may be more accurate and corroborated by the fact that small EBER RNAs were found in all AITL cases.”

This was re-written

Figure legends 6 and 7 lack essential details for comprehension of the data: what the colors mean is not indicated.

The meaning of colors has been added

Hoping that these modifications meet your expectations,

Best regards,

                                                                       Limoges, 09/05/2022

                                                                       Pr S. ROGEZ

Reviewer 2 Report

Epstein-Barr virus (EBV) is a known cause of B and NK/T lymphomas. However, it is not well understood how EBV genetic single nucleotide polymorphisms (SNPs) affect the development of non-virus-infected lymphomas. In this paper, the authors report for the first time that EBV SNPs are strongly associated with Angioimmunoblastic T Lymphoma (AITL). Although this paper proposes a very interesting hypothesis, it is problematic in several important aspects of scientific writing, which need to be corrected.

1) Analysis of EBV-SNPs in the French population

The most important issue in this study is the paucity of healthy human samples: in EBV-positive lymphomas, various mutations are inserted in the viral genome and these mutations are associated with lytic replication (Okuno Y et al., Nat Microbiol, 2019). There is one possibility that the this paper’ suggested SNPs may conversely be repaired in EBV related lymphomas. Therefore, it is unclear whether the EBV SNPs identified by the authors are truly abundant in AITL or whether they are SNPs that are found at high frequency in EBV strains originally infecting the French population. Since only the identified BZLF1 and BRLF1 SNPs are sufficient, it is necessary to analyze SNPs in EBV in healthy individuals. If it is difficult to analyze new healthy samples, data from EBV strains originating in France that are publicly available on the Internet may be used.

2) Notation of figures and tables

Figure Legend needs to describe exactly what is described in the figure. For example, Figure 6 uses red and blue, but there is no description in the Figure legend of what each color represents. Similarly, there is a problem with the description of the figure. What does the vertical axis in Figure 2 indicate? Also, green triangles are recognized in many tables. No explanation is given in the tables.

Moreover, the description states that the A206S SNP in BZLF1 is found in all AITL samples except AIL2, but the figure is different. This difference between description and figure is observed in many figures.

3) Image quality of figures

 In general, figures inserted in scientific papers should be at least 300 dpi. However, the quality of the figures in this paper gives the impression that they are of very low quality and should be prepared at a minimum of 600 dpi.

4) English quality

For example, statements such as Lane 308 are not acceptable in a general scientific paper. English editing is need to improve this papers quality.

5) Discussion

An important aspect of this paper is the discovery of SNPs in EBV that are frequently found in AITL. In this tumor, EBV primarily infects B cells and not AITL. Therefore, what is the difference between the SNPs and mutations in EBV that have been reported in many EBV-positive lymphomas and the mutations in this case? Or are they the same? This needs to be described in Discussion. Multiple references need to be cited on this and compared to data from other groups.

Author Response

Dear Reviewer 2,

Please find attached the revised form of our manuscript ID: cancers- 1699745.

First of all, we would like to thank you for your very constructive comments.

Please find enclosed your comments along with our responses.

1) Analysis of EBV-SNPs in the French population

The most important issue in this study is the paucity of healthy human samples: in EBV-positive lymphomas, various mutations are inserted in the viral genome and these mutations are associated with lytic replication (Okuno Y et al., Nat Microbiol, 2019). There is one possibility that the this paper’ suggested SNPs may conversely be repaired in EBV related lymphomas. Therefore, it is unclear whether the EBV SNPs identified by the authors are truly abundant in AITL or whether they are SNPs that are found at high frequency in EBV strains originally infecting the French population. Since only the identified BZLF1 and BRLF1 SNPs are sufficient, it is necessary to analyze SNPs in EBV in healthy individuals. If it is difficult to analyze new healthy samples, data from EBV strains originating in France that are publicly available on the Internet may be used.

We asked ourselves the question of the existence of mutations in healthy subjects carrying EBV. This is a difficult question to answer.

We collected serum from 30 patients with primary EBV infection. But only 1 sample (the one in the article) had a sufficient viral load to give an NGS result. We examined many lymph nodes from patients for whom a diagnosis of lymphoma had been ruled out. But many of them did not carry EBV or in quantities too low to be studied by NGS. In those that could have been tested by NGS, there was often an inflammatory component that the subject could not be considered "healthy". This is why at the end of all this research we have only one primary infection and one control.Personally, I am not in favor of using the published sequences for which I am not sure that the authors had the same requirements as we did to select the patients. 

2) Notation of figures and tables

Figure Legend needs to describe exactly what is described in the figure. For example, Figure 6 uses red and blue, but there is no description in the Figure legend of what each color represents. Similarly, there is a problem with the description of the figure. What does the vertical axis in Figure 2 indicate? Also, green triangles are recognized in many tables. No explanation is given in the tables.

Figure legends have been revised and corrected.The green triangles on the tables were artifacts and have been removed. 

Moreover, the description states that the A206S SNP in BZLF1 is found in all AITL samples except AIL2, but the figure is different. This difference between description and figure is observed in many figures.

I do not understand the comment because it is written in the text: “Patients AIL2/3/4/5/9/16 harbored the BRLF1 mutation S542N and, except AIL2, the BZLF1 mutation A206S” which corresponds exactly to the figure. 

3) Image quality of figures

 In general, figures inserted in scientific papers should be at least 300 dpi. However, the quality of the figures in this paper gives the impression that they are of very low quality and should be prepared at a minimum of 600 dpi.

I’m sorry, the figures provided at the submission were of good quality, but it was at the time of insertion in the article that they lost their quality. That is why I have reintroduced them all in the text itself. 

4) English quality

For example, statements such as Lane 308 are not acceptable in a general scientific paper. English editing is need to improve this papers quality.

The article has been corrected by a native of the USA. 

5) Discussion

An important aspect of this paper is the discovery of SNPs in EBV that are frequently found in AITL. In this tumor, EBV primarily infects B cells and not AITL. Therefore, what is the difference between the SNPs and mutations in EBV that have been reported in many EBV-positive lymphomas and the mutations in this case? Or are they the same? This needs to be described in Discussion. Multiple references need to be cited on this and compared to data from other groups.

Many papers focus on somatic mutations observed in DLBCLs (for example) associated with EBV. But very few are interested in the mutations of the virus. The mutations reported have mainly been described by their ability to induce proliferation but few have been described in patients. These are discussed in the article.

I hope that these modifications now meet your expectations,

Best regards,

                                                                       Limoges, 09/05/2022

                                                                       Pr S. ROGEZ

Reviewer 3 Report

The authors using whole EBV genome sequencing and other techniques to examine the EBV clonality, latency as well as significant mutations in EBV- associated AITL biopsies versus other EBV- associated lymphomas.  This is a very well written paper with proper controls and statistical analysis. A minor revision will enhance the paper.

  1. Line179-180 “Control1 and PI1 presented heterogeneous EBV strains and cell lines 179 harbored monoclonal strains”,  However, in Fig2, the bar pattern for Control1 and PI1 indicates they belong to Not done category. Please explain what does Not done mean.

  1. Please perform dimension reduction (e.g. PCA or tSNE) based on mutations found in EBV genome to inform whether AITL bears unique EBV mutation clusters when compared to other EBV- associated lymphomas.

Author Response

Dear Reviewer 3,

Please find attached the revised form of our manuscript ID: cancers- 1699745.

First of all, we would like to thank you for your very constructive comments.

Please find enclosed your comments along with our responses.

  1. Line179-180 “Control1 and PI1 presented heterogeneous EBV strains and cell lines 179 harbored monoclonal strains”,  However, in Fig2, the bar pattern for Control1 and PI1 indicates they belong to Not done category. Please explain what does Not done mean.

I think there is confusion between viral clonality and B or T cell clonality, because the figure legend was wrong. Control 1 and PI1 carry heterogeneous strains but B/T clonality was not sought. I edited the caption and I think it is now clearer.

  1. Please perform dimension reduction (e.g. PCA or tSNE) based on mutations found in EBV genome to inform whether AITL bears unique EBV mutation clusters when compared to other EBV- associated lymphomas.

Mutations already described in the literature have been marked in Figure 9 with an asterisk. For the genes in Figure 8, mutations have not been described in the literature.

Hoping that these modifications meet your expectations,

Best regards,

                                                                                   Limoges, 09/05/2022

                                                                                   Pr S. ROGEZ

Round 2

Reviewer 2 Report

The most significant problem with this paper is that it is unclear whether the SNPs identified by the authors are truly clustered in the AITL or whether they are more common in Europe or France. In other words, the problem exists that there is very little data to compare in this paper.

Recently, many genome sequences of EB virus strains have been published. Many of these, for example, can be blast-searched for genetic variants identified by the authors to determine whether these variants are common or truly concentrated in AITL. Many EBVs derived from healthy individuals have been analyzed by infecting cord blood and establishing lymphoblastoid cell lines (LCLs). Therefore, viral strains derived from LCLs are often derived from healthy individuals. LCL analysis is being conducted mainly by research groups in the UK and Germany. Thus, the probability of the presence of data from southern Europe, including France, in these data is very high.

BLAST search should be used to confirm whether the SNPs identified by the authors are indeed enriched in AITL patients.

Next, the papers listed below report next-generation sequencing analyses of large EBV genomes.

  • Okuno Y et al., Nat Microbiol, 2019
  • Xu M et al., Nat Genet, 2019
  • Correia S et al. J Virol, 2018
  • Correia S et al. J Virol, 2017
  • Palser AL et al. J Virol, 2015

Furthermore, SNPs in the viral genome are reported in the following paper.

  • Kim H et al., Biochem Biophys Res Commun, 2019
  • Hui KF et al. Int J Cancer,2019.

Based on these studies, it is necessary to discuss whether these SNPs are really clustered in AITL or not. Although there are only a few examples where the meaning of each SNPs has been clarified, the perspective of viral genomes and SNPs must be written.

Author Response

            Dear Reviewer 2,

Please find attached the revised form of our manuscript ID: cancers- 1699745.

Please find here our response to your comments.

The article entitled “Epstein-Barr virus (EBV) is Mostly Latent and Clonal in
Angioimmunoblastic T cell Lymphoma (AITL)” and submitted for publication in Cancers had the initial objective of reporting the presence of a clonal and latent virus in AITL, as mentioned in the title. It seemed to us that this was the important point and that is why we did not elaborate much on the mutations.

The article has been modified according to the following points:

  • To answer your request on the mutations widespread in our general population, we analyzed the mutations found in LCLs that we established in the laboratory and which concern subjects without hemopathy. Among the mutations found in AITLs and discussed in the article, only 2 mutations on EBNA1 were found.

  • Then, and as you advised, using Blast we searched in published sequences for the presence of each of the mutations discussed in the article. Many mutations were found, sometimes very rarely, and we looked at what type of patient they came from.

  • Finally, we cited the various articles mentioned and reporting complete viral sequences, most of which we were already aware of.

I hope that these modifications now meet your expectations,

Best regards,

                                                                       Limoges, 01/06/2022

                                                                       Pr S. ROGEZ

Round 3

Reviewer 2 Report

All of the reviewers' questions were answered.